# Evaluation of Antioxidant Activity and Biotransformation of *Opuntia Ficus* Fruit: The Effect of In Vitro and Ex Vivo Gut Microbiota Metabolism

**DOI:** 10.3390/molecules27217568

**Published:** 2022-11-04

**Authors:** Ibrahim E. Sallam, Ulrike Rolle-Kampczyk, Stephanie Serena Schäpe, Soumaya S. Zaghloul, Riham S. El-Dine, Ping Shao, Martin von Bergen, Mohamed A. Farag

**Affiliations:** 1Pharmacognosy Department, College of Pharmacy, October University for Modern Sciences and Arts (MSA), 6th of October City, Giza 12566, Egypt; 2Department of Molecular Systems Biology, Helmholtz-Centre for Environmental Research—UFZ GmbH, 04318 Leipzig, Germany; 3Pharmacognosy Department, College of Pharmacy, Cairo University, Cairo 11562, Egypt; 4Department of Food Science and Technology, Zhejiang University of Technology, Hangzhou 310014, China; 5German Centre for Integrative Biodiversity Research, (iDiv) Halle-Jena-Leipzig, Puschstraße 4, 04103 Leipzig, Germany

**Keywords:** *O. ficus-indica*, gut microbiota, biotransformation, UHPLC-QTOF-MS, chemometrics, antioxidant, DPPH, ORAC, FRAP

## Abstract

*Opuntia ficus-indica* biological effects are attributed to several bioactive metabolites. However, these actions could be altered in vivo by biotransformation reactions mainly via gut microbiota. This study assessed gut microbiota effect on the biotransformation of *O. ficus-indica* metabolites both in vitro and ex vivo. Two-time aliquots (0.5 and 24 h) from the in vitro assay were harvested post incubation of *O. ficus-indica* methanol extract with microbial consortium, while untreated and treated samples with fecal bacterial culture from the ex vivo assay were prepared. Metabolites were analyzed using UHPLC-QTOF-MS, with flavonoid glycosides completely hydrolyzed in vitro at 24 h being converted to two major metabolites, 3-(4-hydroxyphenyl)propanoic acid and phloroglucinol, concurrent with an increase in the gallic acid level. In case of the ex vivo assay, detected flavonoid glycosides in untreated sample were completely absent from treated counterpart with few flavonoid aglycones and 3-(4-hydroxyphenyl)propanoic acid in parallel to an increase in piscidic acid. In both assays, fatty and organic acids were completely hydrolyzed being used as energy units for bacterial growth. Chemometric tools were employed revealing malic and (iso)citric acids as the main discriminating metabolites in vitro showing an increased abundance at 0.5 h, whereas in ex vivo assay, (iso)citric, aconitic and mesaconic acids showed an increase at untreated sample. Piscidic acid was a significant marker for the ex vivo treated sample. DPPH, ORAC and FRAP assays were further employed to determine whether these changes could be associated with changes in antioxidant activity, and all assays showed a decline in antioxidant potential post biotransformation.

## 1. Introduction

*Opuntia ficus* is a widely distributed cactus fruit species belonging to the family Cactaceae. *Opuntia* genus contains more than 1500 species found worldwide mainly in Mexico as well as Australia and the Mediterranean region [1]. Many reported pharmacological activities have been attributed to *O. ficus* and its plant constituents including antioxidant [2], anticancer [3], antidiabetic [4] and hepatoprotective [5] activities. Although synthetic analogues of plant bioactives are available and they could exhibit the desired biological activity, plant products are acknowledged to be more effective as the desired activity arises from a cumulative effect of all plant metabolites rather than a single compound [6]. Among the reported bioactives in *O. ficus* include phenolic acids, flavonoids and betalains, to which several studies have attributed the fruit’s biological effects [7,8]. However, in vitro determination of the bioactivity of such metabolites is not sufficient to predict their potential in vivo effects, and this might be attributed to the possible biotransformation that takes place during their interaction with gut microbiota [9]. Therefore, there is an increasing need to establish new approaches to assess not only the biological activity of plant constituents, but also to predict their in vivo effects based on the plant’s chemical composition. Gut microbiota is a large diverse group of microorganisms that live in the gastrointestinal tract, mainly the colon, and their number can reach up to the tens of trillions, which is 10 times higher than the number of human body cells. Bacteria represent the major part of the microbiota with around 1000 species belonging to two main bacterial phyla: *Firmicutes* and *Bacteroidetes* [10]. Many reports have indicated that there is a mutual interaction between the gut microbiota and dietary constituents [11,12,13,14,15,16,17]. As bioactive constituents from dietary substances can influence the composition and the metabolism of gut microbiota, gut microbiota can, on the other hand, yield a series of biotransformed metabolites, thus affecting food biological activity either positively or negatively [18]. Most dietary polyphenols are biotransformed by various enzymes from the gut microbiota inside the colon, and this conversion is essential for their absorption and ultimate biological activity. Accordingly, the colon is regarded as a vital site for metabolism rather than being a simple excretion route [17]. Several biotransformation reactions have been attributed to gut bacterial enzymes including demethylation, dehydroxylation, decarboxylation and ring cleavage, as well as hydrolysis of glycosides, amides and esters [19]. Among the most common effects of gut microbiota on dietary constituents is their impact on plant glycosides, and hydrolysis of such glycosides leads to the formation of metabolites that are potentially more absorbable and thus more biologically active, while further bacterial degradation of aglycones leads to the production of either more or less active compounds based on the metabolites formed [17]. Metabolomics is a holistic approach used for the untargeted high-throughput analysis of complex metabolite matrices that are characteristic of plant extracts [19]. Such an approach was driven by the recent advances in hyphenated techniques such as ultra-performance liquid chromatography coupled to tandem mass spectroscopy (UHPLC-QTOF-MS-MS) analysis. The high-throughput analysis of such a technique represents a vital tool to simplify the complex nature of biotransformation reactions and to aid in monitoring structural changes in an untargeted manner [20]. Multivariate data analysis tools, such as principal component analysis (PCA) and orthogonal projection discriminant analysis (OPLS-DA), have been increasingly applied to help in pinpointing metabolites responsible for discriminating biotransformed extracts, and thus giving new insights on novel compounds with potential biological effect [21]. The aim of this research was to investigate the impact of gut microbiota represented by microbial consortium on the metabolism of plant constituents from *O. ficus* using the UHPLC-QTOF-MS approach in relation to their antioxidant activity aided by chemometric tools.

## 2. Results

### 2.1. In Vitro Impact of Gut Culture Represented by Microbial Consortium

UHPLC-QTOF-MS was employed to monitor various chemical classes of O. ficus metabolites for any possible changes attributed to the effect of the selected microbes. We have previously reported the O. ficus metabolome using same UHPLC-QTOF-MS platform [1], and extend herein to report on how the gut microbiota can impact its metabolite composition. Detected O. ficus fruits’ metabolome is composed mostly of 15 fatty acids, 9 flavonoids, 7 phenolics/phenolic acids, and 7 organic acids, in addition to a triterpenoid (Table 1). Upon inoculation of gut microbiota, dramatic changes in metabolite pattern and percentile levels were detected, among which 6 metabolites were detected post incubation indicating their origin as biotransformed metabolites, being absent from the original fruit matrix. Interestingly, no betalains, which is a major class of metabolites in O. ficus, were detected in both the in vitro and ex vivo assays. This might be attributed to nature of the extraction procedure being pure alcoholic rather than hydroalcoholic, which hinders the extraction of these hydrophilic pigments. It was previously reported that betalains best recovery occurs using pure water extraction and the addition of small percentage of ethanol or methanol, which could enhance their extraction [22]. Likewise, a low operating temperature plays a vital role in maintaining the stability of betalains; it was reported that the most adequate extraction conditions for beetroot betalains occurred with an extraction time of 1 h, operating temperature of 20 °C, and solvent ratio of 0.8 *w/v* of aqueous ethanol, with increased temperature to decreased yield [23]. This also demonstrates that low temperature enhances and preserves the extracted betalains, which was not the case in our research, as O. ficus extract was incubated with bacteria for a period of 24 h at 37 °C, which might have contributed to its degradation. Finally, several studies have reported that betalains’ extraction should be performed in an acidic medium, which is necessary for both the extraction and the preservation of betalains [24], reporting that the optimized conditions for the maximum recovery of betalains were at citric acid of 1.5% and ethanol concentration of 50%. Acidification of the extraction condition might not have favorited the extraction of the other reported classes in O. ficus. Considering that extraction and incubation conditions were not in favor of betalains recovery and/or stability, this explain why no betalains were detected in both assays. A complete list of the identified metabolites with their mass spectral data is presented in Table 1. Chemical structure of major metabolites identified using UHPLC-QTOF-MS is shown in Figure 1.

#### 2.1.1. Flavonoids

Nine flavonoids were detected at 0.5 and 24 h and treated O. ficus within the elution range of t_R_ (400–900 s) (Appendix A), consistent with their nature as relatively non-polar metabolites. The first detected flavonol was isorhamnetin glycoside in **peak 12** with [M-H]^−^ at *m/z* 541.2307 (C_26_H_38_O_12_)^−^, with product ions at *m/z* 315.13 [M-C_9_H_8_O-H]^−^ corresponding to isorhamnetin aglycone [1]. Likewise, **peaks 17** and **19** were detected with molecular formulas of (C_25_H_24_O_15_)^−^ and (C_15_H_10_O_7_^−^) and a common ion at *m/z* 301.03, annotated as quercetin aglycone and quercetin glycoside, respectively [25]. Glycosides denote that the metabolite gave the characteristic product ion of their corresponding aglycone; however, the sugar part was undetermined unequivocally, which is why they were annotated as glycosides in general and not of a specific sugar. Another flavonol was detected in **peak 16** [M-H]^−^ at *m/z* 285.043 (C_15_H_10_O_6_)^−^, with product ions at *m/z* 151.03 [M-C_8_H_6_O_2_-H]^−^ assigned as kaempferol. Naringenin flavanone was detected in **peak 20** with [M-H]^−^ at *m/z* 271.0627 and a molecular formula of (C_15_H_11_O_5_)^-,^ with product ions at *m/z* 151.01 [M-C_8_H_8_O-H]^−^ and *m/z* 119.04 [M-C_7_H_4_O_4_-H]^−^. Another flavonoid subclass is that of flavones, which were detected in **peaks 26, 27** and **28** corresponding to apigenin [M-H]^−^
*m/z* 269.0478 (C_15_H_9_O_5_)^−^, diosmetin [M-H]^−^ at *m/z* 299.0577 (C_16_H_12_O_6_)^−^ and acaetin [M-H]^−^ *m/z* 283.0643 (C_16_H_12_O_5_)^−^ aglycones. Confirmation of aglycone was based on product ions at *m/z* 151 [M-C_8_H_4_O-H]^−^ and *m/z* 117.03 [M-C_7_H_4_O_4_-H]^−^ in **peak 26** versus *m/z* 284.03 [M-CH_3_-H]^−^ and *m/z* 151 [M-C_8_H_6_O_2_-H]^−^ [26] in **peak 27** and *m/z* 151.01 [M-C_9_H_8_O-H]^−^ and *m/z* 117.03 [M-C_8_H_6_O_4_-H]^−^ in **peak 28 [27]**. 

##### Biotransformation of Flavonoids

Almost all bacterial strains employed within the ex vivo assay are reported to code for hydrolytic enzymes, mainly β-glucosidase. These enzymes are reported to be active within few minutes of bacterial incubation with plant extracts viz., Bacteroides thetaiotaomicron [28], Bifidobacterium longum [15], Clostridium genus [29], Escherichia coli [30] and Lactobacillus plantarum [31]. Thus, most of the detected flavonoids were aglycones at both the initial 0.5 h and final 24 h time points, suggesting that they were readily metabolized to their respective aglycone and contrary to the abundance of glycosides in the fruit extract [32]. Interestingly, a differential response among flavonoid glycosides was observed exemplified in the rapid degradation of flavone/flavonones compared to flavonols. Flavonoid aglycones, such as naringenin, diosmetin and acacetin, were only detected at 0.5 h and completely hydrolyzed at 24 h, whereas quercetin and isorhamnetin flavonol glycosides were detected at 0.5 and 24 h, with 0.48- and 0.82-fold decrease in their levels, respectively, at the late time point. The detection of kaempferol aglycone at 24 h and absent from 0.5 h suggests that it is a hydrolytic product from the bacterial metabolism of plant constituents. As a result of these differential responses, the total flavonoid content showed relatively similar percentages between the 0.5 and 24 h samples (Figure 2A, Appendix A).

Following aglycone cleavage from glycoside, it undergoes extensive metabolism by the colon bacteria to simpler phenolics from A and B rings mediated by the C-ring cleavage (Figure 3). Clostridium [16], Eubacterium [33] and other gut microbiota belonging to the Butyrivibrio genus [34], employed herein, are among the bacterial strains reported to be involved in such cleavage. Subsequent to the C ring cleavage, dehydroxylation occurs with the original hydroxylation of A and B rings found to affect the resulting metabolites. A major metabolite suggested to be derived from the dehydroxylation of the B ring is 3-(4-hydroxyphenyl) propionic acid (Figure 3), detected exclusively at 24 h in **peak 15** with [M-H]^−^ *m/z* 165.0578 (C_9_H_9_O_3_)^−^, with product ions at *m/z* 147.04 [M-H_2_O-H]^−^ and *m/z* 119.05 [M-H_2_O-CO-H]^−^ (Appendix A) [16,33]. Likewise, the phloroglucinol A ring cleavage product was detected only at 24 h in **peak 7** with [M-H]^−^ *m/z* 125.0256 (C_6_H_5_O_3_)^−^, with product ions at *m/z* 106.14 [M-H_2_O-2H]^−^ and *m/z* 91.08 [M-2OH-H]^−^ (Appendix A) [34]. Phloroglucinol is a polyphenolic compound with a broad range of reported biological effects including antioxidant [35], cytotoxic [36] and antidiabetic [37] activities. Moreover, it has been widely used for the treatment of spasmodic pain associated with irritable bowel syndrome and renal colic [38].

#### 2.1.2. Phenolics and Organic Acids

Phenolic and organic acids were the second most abundant classes represented by 14 metabolites. Their abundance is visible within the elution region of t_R_ (50–500 s) (Appendix A), being most polar and eluting at a high water eluent composition. Among the detected organic acids were malic acid in **peak 1** with [M-H]^−^ at *m/z* 133.0160 (C_4_H_6_O_5_)^−^, with product ions at *m/z* 115 [M-H_2_O-H]^−^ and *m/z* 71.01 [M-H_2_O-COO-H]^−^ (Appendix A), and succinic acid in **peak 6** with [M-H]^−^ at *m/z* 117.0205 (C_4_H_6_O_4_)^−^, with product ions at *m/z* 99.01 [M-H_2_O-H]^−^ and *m/z* 73.03 [M-COO-H]^−^ [39]. Several peaks were annotated for (iso)citric acid and its derivatives in **peak 3** [M-H]^−^ at *m/z* 191.0222 (C_6_H_7_O_7_)^−^ with product ions at *m/z* 173.03 [M-H_2_O-H]^−^ and *m/z* 129.01 [M-COO-H_2_O-H]^−^ annotated as (iso)citric acid (Appendix A), and its derivatives in **peaks 4, 8** and **13** with similar fragmentation pattern showing product ions at *m/z* 191.05 [M-R-H]^−^ and *m/z* 127 [M-R-COO-H_2_O-H]^−^ ascribed for the presence of (iso)citric acid moiety and annotated as hydroxy and homocitric acid and dimethyl citrate, respectively [40]. Finally, **peak 10** was assigned as fumarylacetoacetic acid [M-H]^−^ at *m/z* 199.0265 (C_8_H_7_O_4_)^−^ with a product ion at *m/z* 101.38 ascribed for the loss of fumaric acid [21]. Alternatively, gallic acid was detected in **peak 2** [M-H]^−^ at *m/z* 169.0158, with product ions at *m/z* 125.02 [M-COO-H]^−^ and *m/z* 107.01 [M-COO-H_2_O-H]^−^ (Appendix A). Other gallic acid derivatives were detected in **peaks 14** and **18** with product ions at *m/z* 169.01 [M-R-H]^−^, *m/z* 125.02 [M-R-COO-H]^−^ and *m/z* 107.01 [M-R-COO-H_2_O-H]^−^ ascribed for gallic acid moiety being assigned as methyl gallate and ethyl gallate esters, respectively [41].

##### Biotransformation of Phenolics and Organic Acids

Phenolics showed a 2.5-fold increase, while total organic acids showed an 0.82-fold decrease post incubation (Figure 2A, Appendix A). The most pronounced decrease in organic acids was observed in case of malic and (iso)citric acids and its derivatives by 0.2- and 0.8-fold, respectively. These organic acids are likely utilized within the glyoxylate pathway (Figure 4) for metabolic energy production necessary for bacterial growth in culture [42]. In contrast, succinic acid, a structural analogue to malic acid, showed an opposite pattern being found at increased levels at 24 h (ca. 1.72-fold), which is likely attributed to the highly reported microbial production of succinic acid from (iso)citric acid through isocitrate lyase as a major fermentation product [43]. Fumarlyacetoacetic acid (maleylacetoacetic acid) showed a similar accumulation pattern (ca. 3.8-fold increase), which is attributed to being an intermediate product in the metabolism of the tyrosine amino acid mainly by fumarylacetoacetate hydrolase, which is necessary for its use as a substrate in energy production [44].

With regards to gallic acid derivatives showing more abundance at 0.5 h versus gallic acid detected at higher levels at 24 h (1.5-fold increase), this indicates the hydrolysis of these derivatives via esterase (tannase) and or decarboxylase enzymes. Another hydrolytic product of phenolics detected in culture was pyrogallol found exclusively at 24 h as a possible hydrolytic product of gallic acid decarboxylation. Lactobacillus plantarum included in the culture consortium is the only reported bacterial species to encompass esterase (tannase) and decarboxylase enzymes [12], which are essential for the bacterial hydrolysis of gallotannins abundant in O. ficus [45]. Gallic acids and their decarboxylated metabolite pyrogallol exhibit several pharmacological effects including antioxidant [46], antimicrobial [47] and anticancer [14] activities.

#### 2.1.3. Fatty Acids

Fatty acids were the most abundant metabolite class in extract represented by 15 peaks as evident from the total ion chromatogram within the elution range of t_R_ (750–1450 s) (Appendix A). The late elution is consistent with the non-polar nature of these metabolites to include linoleic and oleic acids detected in **peaks 37** and **39** with [M-H]^−^ at *m/z* 279.2351 and 281.2521 and a molecular formula of (C_18_H_31_O_2_)^−^and (C_18_H_33_O_2_)^−^, respectively. Two mono-hydroxylated fatty acids were detected in **peaks 32** and **34** showing loss of water molecule (-18 amu) and annotated as hydroxylinoleic acid and hydroxypalmitic acid, respectively; likewise, palmitoleic acid and palmitic acid were detected in **peaks 36** and **38,** respectively [48,49]. **Peak 23** with [M-H]^−^ at *m/z* 329.2358 (C_18_H_32_O_5_)^−^ with product ions at *m/z* 155.03 [M-C_7_H_10_O_5_-H]^−^ and *m/z* 133.01 [M-C_14_H_28_-H]^−^ presented a typical fragmentation pattern of tri-hydroxyoctadecenoic acid [20]. Another dihydroxy fatty acid was detected in **peak 21** [M-H]^−^ at *m/z* 287.2249 (C_16_H_31_O_4_)^−^ with product ions at *m/z* 271.02 [M-O-H]^−^ and *m/z* 243.05 [M- COO-H]^−^ assigned as dihydroxyhexadecanoic acid. A similar fragmentation pattern was observed in **peaks 22, 24** and **25,** assigned as tri- and dihydroxy fatty acid derivatives.

##### Biotransformation of Fatty Acids

Upon bacterial inoculation, fatty acids showed a decline of 0.6-fold (Figure 2A, Appendix A), as almost all detected fatty acids were depleted at 24 h, which is attributed to the fact that all employed bacterial strains within this assay are reported to metabolize long-chain fatty acids (LCFAs) to short-chain fatty acids (SCFAs) i.e., acetate, propionate, butyrate and valerate (Figure 4). This was evident by the post incubation 1.75-fold increase of valerate SCFA observed in **peak 11** as hydroxyvaleric acid with [M-H]^−^ at *m/z* 117.0566 and a molecular formula of C_5_H_9_O_3_^−^ [50]. Valerate is exclusively produced by bacterial fermentation of amino acids and polypeptides, as well as fatty acids produced by several bacterial species including Clostridium, which is used in this assay [51]. Therefore, gut microbiota composition affects the production of SCFAs, which have been linked to improving the gut health through a number of local effects, ranging from maintaining intestinal barrier integrity, mucus production and protection against inflammation to reduction of the risk of colorectal cancer [52]. Fatty acids are considered a major energy source, whereas their derivatives regulate several cellular responses essential for bacterial growth [13]. Fatty acids along with carbohydrates metabolism are mediated anaerobically via the glyoxylate pathway that leads to the formation of SCFAs (Figure 4) [43].

### 2.2. Impact of Gut Culture Represented by Actual Fecal Matter in Ex Vivo Assay

To better assess of gut microbiota on O. ficus fruit metabolites, methanol extract of O. ficus was incubated with an ex vivo culture of the human gut microbiome isolated from fecal matter of a healthy donor as mentioned under Experimental Section 2.4. The longer incubation time of 48 h, as well as utilizing actual fecal matter as a source for gut microbiota, yielded better insight on the possible biotransformation reactions that occur within the human body and helped confirm results derived from in vitro gut culture assay (Figure 2A). The same UHPLC-QTOF-MS-MS platform was employed to detect metabolite changes between the untreated and treated O. ficus extract. A representative chromatogram of O. ficus untreated and treated samples is depicted in Appendix A. Similar metabolite classes as the in vitro assay were detected, including 14 phenolic and organic acids, 9 flavonoids and 10 fatty acids. Likewise, dramatic changes were observed in metabolites, including 6 peaks detected post incubation in the treated sample, indicating their origin as biotransformed metabolites rather than being originally present within O. ficus matrix. A complete list of the identified metabolites with their mass spectral data is presented in Appendix A. 

#### 2.2.1. Biotransformation of Flavonoids

In contrast to the 0.5 h sample employed in the in vitro assay, all detected flavonoids in the untreated fruit extract in case of the ex vivo culture were glycosides, confirming the abundance of flavonoids as glycosides within the native plant matrix (Appendix A), as well as rapid hydrolytic effect of bacterial glucosidase enzymes in the case of the in vitro culture. Upon incubation, all flavonoid glycosides, in spite of their classes, were completely hydrolyzed at 48 h, at which point only their respective aglycones are a product of bacterial metabolism (Appendix A). Accordingly, the total flavonoids percentage remained relatively the same within the untreated and treated samples (Figure 2B). As in the in vitro assay, 3-(4-hydroxyphenyl) propanoic acid was also detected exclusively at 48 h in the treated sample in **peak 17** as a possible hydrolytic product of flavonoid aglycone cleavage imparted by bacterial metabolism (Figure 3).

#### 2.2.2. Phenolic and Organic Acids

In the ex vivo incubation assay, phenolic and organic acids were the most abundant class with 15 metabolites. Their abundance is visible within the total ion chromatogram at the elution region of t_R_ (50–530 s) (Appendix A), consistent with their high polarity. Organic acids other than those detected within the in vitro assay were detected including gluconic acid in **peak 1** with [M-H]^−^ at *m/z* 195.0504 (C_6_H_11_O_7_)^−^, with product ions at *m/z* 177.01 [M-H_2_O-H]^−^ and *m/z* 133.03 [M -COO-H]^−^ [1]. **Peak 8** was detected with [M-H]^−^ at *m/z* 173.0091 (C_6_H_5_O_6_)^−^ with product ions at *m/z* 129.02 [M-COO-H] and *m/z* 111.01 [M-H_2_O-COO-H]^−^, which is a typical fragmentation pattern of aconitic acid (Appendix A). Mesaconic acid was detected in **peak 9** with [M-H]^−^ at *m/z* 129.0216 (C_5_H_5_O_4_)^−^ with product ion at *m/z* 85.05 [M-COO-H]^−^ (Appendix A) [53]. Likewise, phenolic acids including caffeoylquinic acid were detected in **peak 2** with [M-H]^−^ at *m/z* 353.0862 with a molecular formula of C_16_H_17_O_9_^−^, with product ions at *m/z* 191.01 [M-179-H]^−^ [54]. Cinnamic acid was detected in **peak 10** with [M-H]^−^ at *m/z* 147.0454 (C_9_H_7_O_2_)^−^, with product ions at *m/z* 129.01 [M-H_2_O-H]^−^ and *m/z* 85.01 [M-H_2_O-COO-H]^−^. Another phenolic acid was detected in **peak 11** with [M-H]^−^ at *m/z* 255.0557 with a molecular formula of C_11_H_11_O_7_^−^, with product ions at *m/z* 165.05 [M-C_2_H_2_O_3_-OH-H]^−^, *m/z* 119.05 [M-C_2_H_2_O_3_-OH-CH_2_-H]^−^ and 107.05 [M-C_4_H_4_O_3_-H]^−^, and it was annotated as piscidic acid (Appendix A) [55]. Galloylhexose was detected in **peak 14** with [M-H]^−^ at *m/z* 331.0681 with a molecular formula of C_13_H_15_O_10_^−^, with product ions at *m/z* 169.01 [M-C_6_H_10_O_5_-H]^−^ ascribed for the loss of glucose moiety. A few metabolites in **peaks 18, 21** and **25** showed a similar fragmentation pattern with product ions at *m/z* 331.06 [M-R-H]^−^ and 169.03 [M-R-C_6_H_10_O_5_^−^H]^−^, suggesting that these metabolites are galloylhexose derivatives [56]. 

##### Biotransformation of Phenolic and Organic Acids

Upon incubation, organic acids showed a similar biotransformation behavior to the in vitro assay with a 0.58-fold decrease in treated sample compared with increase in phenolic acids at 4-fold when compared to the untreated sample at 48 h (Figure 2B, Appendix A). Almost all organic acids were completely depleted upon incubation, except for succinic acid detected at 48 h, confirming its origin as a bacterial metabolite [43]. This can also be attributed to organic acids’ utilization in the aforementioned glyoxylate pathway for energy production with subsequent production of succinic acid through isocitrate lyase as a major fermentation product [42,43]. As for phenolic acids, the most pronounced increase was observed in case of piscidic acid, showing a 12-fold increase at 48 h, which is likely attributed to the release of piscidic acid and other polyphenols from their partial binding to the dietary fibers present in O. ficus under the impact of bacterial enzymes [57]. The detection of gallic acid only at 48 h indicates its absence from the original plant matrix, whereas its detection within the 0.5 h sample in the in vitro assay indicates that, like flavonoids, the 0.5 h treatment was sufficient to release gallic acid from its derivatives (esters). This can be further confirmed by the detection of galloylhexose and its derivatives in the untreated sample of the ex vivo assay (Appendix A) with no evidence for their presence in the 0.5 h sample of the in vitro assay (Table 1). These galloylhexose derivatives [58] were completely depleted upon incubation, indicating their usage as substrates for bacterial growth. Likewise, caffeoylquinic acid was depleted with incubation, while cinnamic acid showed a 0.3-fold decrease. 

#### 2.2.3. Fatty Acids 

Next to organic and phenolic acids, fatty acids were the second most abundant class with 10 metabolites visible within the elution range of t_R_ (710–1370 s) (Appendix A), which is consistent with their non-polar nature. All of these metabolites were those detected in the in vitro assay except for a fatty acid, which was detected in **peak 29** with [M-H]^−^ at *m/z* 235.1736 and a molecular formula of (C_15_H_24_O_2_)^−^, with a product ion at *m/z* 217.17 ascribed for the loss of water molecule (−18 amu) and annotated as trimethyldodecatrienoic acid (farnesoic acid) [59].

##### Biotransformation of Fatty Acids

As in the in vitro assay, total fatty acids showed a decline of 0.6-fold (Figure 2B, Appendix A) as almost all fatty acids were depleted upon treatment at 48 h due to the reported bacterial metabolism of LCFAs to SCFAs. This was evident from the detection of propionic acid in **peak 3** with [M-H]^−^ at *m/z* 73.0311 and (C_3_H_5_O_2_)^−^ exclusively at 48 h, as well as the post incubation 8-fold increase of the previously described hydroxyvaleric acid in **peak 12**. Propionate SCFA is mainly produced by Bacteroidetes phyla through fermentation of carbohydrates as well as organic, fatty and amino acids [60], and to possess an antimicrobial activity against the colonization of the GIT with pathogenic bacteria, such as Salmonella typhimurium, through inhibiting their invasion genes that are essential for penetrating the intestinal epithelium [61]. Additionally, propionic acid intake has been reported to exert a beneficial effect on insulin sensitivity modulated by it inhibitory effect on free fatty acids’ metabolism as well as inflammation associated with insulin resistance [62].

### 2.3. Multivariate Data Analysis of Fermented O. ficus Extracts Using In Vitro and Ex Vivo Cultures

Multivariate data analysis of the UHPLC-QTOF-MS of the two time points, 0.5 and 48 h, was used to determine metabolic markers for microbial fermentation in an untargeted manner. The UHPLC-QTOF-MS peak abundance-extracted dataset of the in vitro and ex vivo assays were both subjected to unsupervised PCA and supervised OPLS modelling to identify biomarkers for each time point. 

#### 2.3.1. Multivariate Data Analysis of MS Dataset of O. ficus in Response to In Vitro Gut Bacterial Culture

The unsupervised data analysis (PCA) failed to provide clear segregation between the two time aliquots in response to the treatment; thus, supervised orthogonal partial least squares discriminant analysis (OPLS-DA) was further employed as supervised model by pooling samples at 0.5 h in one class group versus 24 h in another class group (Figure 5A). Model validation was based on estimating the total variance (R2 = 0.97), prediction goodness parameter (Q2 = 0.92) and *p*-value for statistical significance. The OPLS model showed clear samples’ segregation with high repeatability, prediction and significantly low regression *p*-values that are suggestive of no model overfitting. The S-loading plot was further investigated to visualize both the covariance and the correlation structure between the X-variables and the predictive score (1) of the model. Whereas metabolites with negative *p*(1) values indicate higher abundance at 0.5 h and a decrease upon incubation, positive *p*(1) values indicate an increase upon incubation. (Iso)citric as well as malic acid were the only significant model markers for the 0.5 h time point harvest with negative *p*(1) on S-plot (Figure 5B), likely due to their incorporation and consumption within the glyoxylate pathway needed for energy production. In contrast, no significant markers could be assigned for the 24 h time point.

#### 2.3.2. Multivariate Data Analysis of MS Dataset of *O. ficus* in Response to Ex Vivo Fecal Bacterial Culture

As in the in vitro assay, PCA failed to provide clear segregation between the untreated and treated samples; thus, OPLS-DA was further employed by pooling the untreated samples in one class group versus treated samples in another class group (Appendix A). Model validation was based on estimating the total variance (R2 = 0.99) and the prediction goodness parameter (Q2 = 0.92). The OPLS model showed samples segregation, and (iso)citric, aconitic and mesaconic acids were the only significant model markers for the untreated sample (positive *p*(1) values). In contrast, piscidic acid was a significant marker for the treated sample at 48 h (negative *p*(1) value), which might be attributed to polyphenols’ release from their partial binding to O. ficus dietary fibers under the impact of bacterial enzymes.

### 2.4. Antioxidant Effect of Inoculated O. ficus Samples

Free radicals play an important role in the progression of oxidative stress and many other associated diseases, i.e., cancer and cardiovascular disorders [63]. Thus, there is an increasing interest in not only developing new antioxidant principles especially from plant origin, but also in determining their fate and metabolism inside the body. To assess whether bacterial inoculation and gut culture-mediated biotransformation influence the O. ficus antioxidant effect, we determined extract effects for cultures harvested at 0.5 and 24 h in vitro using 2,2-Diphenyl-1-Picrylhydrazyl (DPPH), Ferric-Reducing Antioxidant Power (FRAP) and Oxygen Radical Absorbance Capacity (ORAC) assays. The 0.5 h time point showed higher antioxidant activity in both ORAC and FRAP assays (137.5 and 22.3 µM TE/mg extract, respectively), compared to 24 h aliquot (40.8 and 7.9 µM TE/mg extract, respectively). Likewise, in the DPPH assay, the two time aliquots (0.5 and 24 h) in the in vitro assay showed slightly higher IC_50_ values of 191.2 µg/mL at 24 h compared to 174.1 µg/mL at 0.5 h and compared to that of the standard trolox (24.42 µg/mL). The higher antioxidant capacity demonstrated by original fruit extract subjected to less microbial degradation (0.5 h aliquot) suggests that the microbial degradation and metabolites’ biotransformation exhibited in the 24 h aliquot negatively influence its antioxidant activity. Whether the decline in other effects attributed to O. ficus follow the same pattern should be examined.

## 3. Materials and Methods

### 3.1. Plant Material

Fresh samples from red ‘Rose’ cultivar of *O. ficus* were harvested when fully mature ranging in size from 4 to 9 cm (length) and 3 to 5 cm (diameter). Fruits were peeled using a razor blade and were lyophilized as whole parts using a Stellar^®^ Laboratory Freeze Dryer (Millrock, Inc., New York, NY, USA), stored at −20 °C and extracted after grinding within 1–2 wk for metabolite analysis. Methanol extract was prepared from peeled *O. ficus* ‘Rose’ FI fruit powder by cold maceration over 48 h using 100% methanol until exhaustion. Extract was then filtered, and the supernatant was subjected to evaporation under vacuum at 40 °C until complete dryness. Extracts were placed in tight glass vials and stored at −20 °C until further analysis.

### 3.2. Gut Microbiota Culture

The microorganism consortium used in this study is a model for the intestinal microbiota and described as the extended simplified intestinal human microbiota—SIHUMIx. Microorganisms were selected according to their occurrence in humans, the spectrum of fermentation products formed and the ability to form a stable community by [64]. Co-cultured bacterial species included *Anaerostipes caccae* (DSMZ 14662), *Bacteroides thetaiotaomicron* (DSMZ 2079), *Bifidobacterium longum* (NCC 2705), *Blautia producta* (DSMZ 2950), *Clostridium butyricum* (DSMZ 10702), *Clostridium ramosum* (DSMZ 1402), *Escherichia coli K-12* (MG1655) and *Lactobacillus plantarum* (DSMZ 20174), all cultivated as single and provided by the Helmholtz-Centre for Environmental Research-UFZ, Leipzig, Germany. Culturing of bacteria was previously described in [19]. Briefly, all bacteria were cultivated in brain–heart infusion (BHI) medium (Roth^®^, Karlsruhe, Germany) under anaerobic conditions at 37 °C and 175 rpm and shaken for 72 h prior to inoculation. All strains were shown to be able to grow equally in the medium. The BHI medium was prepared by mixing 37 g of brain–heart infusion, 0.5 g of L-cysteine hydrochloride (Biochemica^®^, Ulm, Germany), 0.001 g of resazurin (MP biomedicals, Irvine, CA, USA), 10 mL of Vitamin K hemin solution (Becton Dickinson, Sandy, UT, USA) and 5 g of yeast extract (Chemsolute, Renningen, Germany) in 1 L of sterile water. Gut bacteria cultured in the brain–heart infusion medium (optical density of 0.1, measured at 600 nm) was left to grow under anaerobic condition at 37 °C for 18 h until the optical density reached 1.7 prior to *O. ficus* extract addition. Each culture was performed in triplicate to assess for biological replicates for each treatment, in addition to 3 blank cultures made of SIHUMI and BHI medium without treatment.

### 3.3. In Vitro Incubation of Plant Extract with Gut Bacterial Culture

Stock solution of *O. ficus* was prepared at an initial concentration of 50 mg/mL in 50:50 methanol: growth media. First, a 1 mL aliquot of the stock solution was incubated in 10 mL of growth media containing selected microbes from the gut microbiota to achieve a final concentration of 5 mg/mL. Finally, 3 to 4 mL of the prepared samples were harvested at two time intervals, 0.5 and 24 h, for analysis to represent the initial time point, at which no biotransformation reactions are expected to have occurred, and the final time point, at which all biotransformation reactions would have occurred by then microbiota-treated functional food assays, respectively. This method was previously reported by our group to assess the mutual impact between gut microbiota and seven functional foods regarding their primary metabolites [19]. Blank cultures were prepared by adding an equivalent amount of 50 and 500 µL 100% methanol into the culture medium, kept under the same condition and compared to the culture receiving no solvent treatment. All harvested aliquots as well as blanks were subjected to UHPLC-QTOF-MS-MS analysis to monitor the chemical changes of different classes of plant metabolites attributed to the selected gut microbes. All analyses were done from three independent triplicate cultures.

### 3.4. Ex Vivo Incubation of Plant Extract with Gut Culture from Donor Fecal Sample

Human subject fecal sample was cultured at Princeton University Department of Molecular Biology and provided in this experiment. The exact experimental details of fecal sample collection from the pilot donor (PD), storage, ex vivo culture and xenobiotic incubation were recently described in [65]. All ex vivo cultures in this study were done with PD in mGAM medium (HyServe^®^, Uffing, Germany) under anaerobic conditions in a water bath at 85 °C using CO_2_ stream. Resazurin dye (1 g/L) was used as an anaerobic indicator, and 1 mL of sodium sulfide (Sigma-Aldrich, Taufkirchen, Germany) solution (240 g/L) was added to 1 L of anaerobic medium to originally change the resazurin color to yellow (reduced form); samples that turned pink during incubation were discarded for oxygen infiltration. Then, 10 mg of *O. ficus* extract was incubated with 1 mL mGAM containing the ex vivo culture in an anaerobic jar containing AnaeroPack^®^ (Mitsubishi Gas Chemical America, Inc., New York, N.Y.) and then incubated in a shaking incubator at 37 °C for 48 h. Another 10 mg of the plant extract was prepared under the same conditions and incubated for the same period without the ex vivo culture to represent the untreated plant extract. All analyses were done from three independent triplicate cultures.

### 3.5. Metabolites Extraction and UHPLC-QTOF-MS-MS Analysis

First, 200 µL of harvested cultures from both in vitro and ex vivo assays was spiked with umbelliferone (Sigma-Aldrich, Taufkirchen, Germany) standard solution dissolved in sterile water to reach a final concentration of 10 µg/mL followed by the addition of 800 µL acetonitrile/methanol mixture (1:1) with incubation at 4 °C for 30 min until complete protein precipitation. The mixture was then centrifuged at 12,000 g using Eppendorf centrifuge for 4 min, with 100 µL of the supernatant then aliquoted and subjected to high-resolution UHPLC-QTOF-MS-MS analysis. For LC-MS/MS measurement, 10 µL of each extract was injected onto a HPLC system coupled online with a 6540 UHD Accurate-Mass Q-TOF (Agilent Technologies). Separation was achieved on a Waters Acquity UPLC^®^ CSH C18 column (2.1 × 100mm, 1.7µm) equipped with a Waters Acquity UPLC^®^ CSH C18 pre-column (2.1 × 50mm, 1.7µm). The autosampler was kept at 5 °C and the column oven was set to 45 °C. Metabolites were separated using a binary solvent system (A: 0.1% FA in water and B: 0.1% FA in ACN) running with the following gradient: 0–5 min: 5% B; 5–19 min: 5–95% B; 19–21 min: 100% B; 21–21.5: 100–5% B; and 21.5–24 min: 5% B. Metabolites were eluted at a constant flow rate of 0.3 mL/min. QTOF was operated in centroid mode and full-scan data were generated with a scan range of 60–1600*m/z* in positive and negative ionization mode. Out of the survey scan, the 5 most abundant precursor ions with charge state = 1 were subjected to fragmentation. The dynamic exclusion time after two acquired spectra was set to 30 s. The used chromatographic conditions have been successfully used for profiling similar plant matrices [66,67,68].

### 3.6. UHPLC-QTOF-MS-MS Multivariate Data Analyses

MS peak abundance of metabolites were extracted using MS-DIAL version 4.6 (RIKEN, Yokohama, Japan) as previously described in [69]. The aligned peak abundance data table was further exported to principal component analysis (PCA) and orthogonal projection least squares discriminant analysis (OPLS-DA) using SIMCA-P version 14.1 software package (Umetrics, Umeå, Sweden). All variables were mean-centered and scaled to Pareto variance (Par).

### 3.7. Antioxidant Assays of Inocculated O. Ficus Samples

#### 3.7.1. DPPH Antioxidant Assay

The antioxidant capacity was determined by the scavenging DPPH radical (Cayman Chemical, Ann Arbor, MI, USA) as described in [1]. Each time aliquot (50 *µ*L) was mixed with 2 mL of 0.09 mM DPPH solution using a shaker at 25 °C and 1000 rpm, followed by incubation at room temperature for 15 min in the dark. Absorbance was measured at 517 nm using a Beckman Coulter DTX 880 microplate reader (Biodirect Corp., Taunton, MA, USA). Trolox (Sigma-Aldrich, Taufkirchen, Germany) was used as a positive control initially prepared as a stock solution of 100 µM in methanol, from which 7 serial dilutions were prepared including 5, 10, 15, 20, 30, 40 and 50 µM. Blank samples were prepared by replacing the time aliquot with 100% methanol. Triplicates were done for each measurement prepared from a different specimen using the same conditions, and the results are expressed as IC_50_.

#### 3.7.2. FRAP Antioxidant Assay

The ferric-reducing ability assay was carried out according to the method of [70] with slight modifications to be carried out in microplates. Briefly, a freshly prepared tripyridyltriazine (TPTZ) reagent (Sigma-Aldrich, Taufkirchen, Germany) (300 mM Acetate Buffer (PH = 3.6), 10 mM TPTZ in 40mM HCl and 20 mM FeCl_3_, in a ratio of 10:1:1 v/v/v, respectively) was used. First, 190 µL of the freshly prepared TPTZ reagent were mixed with 10 µL of the sample in 96-well plates (*n* = 3), and the reaction was incubated at room temperature for 30 min in the dark. At the end of incubation time, the resulting blue color was measured at 593 nm. Trolox stock solution of 2 mM in methanol was prepared, and 8 serial dilutions were prepared in the concentrations of 50, 100, 200, 400, 600, 800, 1000 and 1500 µM. Data are represented as means ± SD. The ferric-reducing ability of the samples is presented as µM trolox equivalent (TE)/mg sample using the linear regression equation extracted from the linear dose–response curve of Trolox.

#### 3.7.3. ORAC Antioxidant Assay

The assay was carried out according to the method of [71], with minor modifications; briefly, 12.5 µL of the prepared samples were incubated with 75 µL fluoresceine (Sigma-Aldrich, Taufkirchen, Germany) (10 nM) for 30 min at 37 °C. Fluorescence measurement (485 EX, 520 EM, nm) was carried out for three cycles (cycle time, 90 sec.) for background measurement. Afterward, 12.5 µL of freshly prepared 2,2-azobis(2-amidinopropane (AAPH) (Sigma-Aldrich, Taufkirchen, Germany) (240 mM) were added immediately to each well. Fluorescence measurement (485 EX, 520 EM nm) was continued for 2.5 h (100 cycles, each 90 s). Trolox stock solution of 1mM in methanol was prepared, and 9 serial dilutions were prepared in the concentrations of 400, 300, 200, 150, 100, 75, 50, 25 and 12.5 µM. Data are represented as means (*n* = 3) ± SD, and the antioxidant effect of the compound/extract was calculated as µM Trolox equivalents by substitution in the linear regression equation.

Figure 6 illustrates a schematic diagram of the experimental workflow used in this assay.

## 4. Conclusions

*O. ficus* is a widespread functional food with many reported biological activities attributed mostly for its fruit’s richness in phenolics. Biotransformation of *O. ficus* bioactive metabolites is a critical detrimental factor of their absorption, and thus their pharmacological effects, especially in the colon where they are produced. Our results demonstrate the biotransformation pathway adopted by the gut microbiota for the metabolism of flavonoid glycosides in *O. ficus* fruits through rapid release of their respective aglycones, which are considered more readily absorbable, and hence, more biologically active [72]. Nevertheless, further bacterial degradation of flavonoid aglycones leads to the generation of other derivatives such as phloroglucinol. Polyphenolics, which like flavonoids are considered rich metabolites of *O. ficus* fruit, were found to be metabolized through the formation of simpler phenolic compounds, i.e., gallic acid and piscidic acids as well as pyrogallol. It should be noted that the current study did not assess the effect of gut microbiota on betalains’ metabolism considering their presence at trace levels in the 100% alcohol extract prepared from the fruits. Future studies can target this class by extracting fruits using polar hydroalcoholic or aqueous solvents. This study showed the bacterial utilization of fatty and organic acids for the production of metabolic energy through glyoxylate pathway evident from their decrease post incubation. Bacterial metabolism of fatty acids was shown to be mediated via the production of SCFAs, presenting an added value considering their role in many metabolic and inflammatory diseases. Moreover, both the in vitro and ex vivo assays showed the same biotransformation effect regarding metabolite classes, i.e., flavonoids, fatty acids and organic acids, with the main mentioned differences being a factor of time. As in the ex vivo assay, we incubated the completely untreated vs treated plant sample for 48 h, while in the in vitro assay, we used 0.5 h vs 24 h aliquots. This demonstrates that the incubation time of gut microbiota with plant metabolites is the main detrimental factor affecting their bioavailability and bioaccessibility. Furthermore, an in vivo approach through human ingestion of plant or food products followed by collection of fecal samples for chemical analysis will not only give more insights on the effect of gut microbiota on plant constituents, but would also give better understanding of the gastrointestinal tract effect as a whole. Likewise, this study demonstrated the impact of gut microbiota on *O. ficus* antioxidant activity to suggest that biotransformation lessens antioxidant activity according to the DPPH, FRAP and ORAC assays. Determination of other biological effects using isolated biotransformed compounds of bacterial origin should now be conclusive about their importance on the antioxidant activity as well as on others. It should be noted that the adopted metabolites’ extraction conditions and further incubation assay were not in favor of recovering betalains, a pigment in Opuntia fruits. Betalains are polar pigments that need an aqueous solvent for their recovery at acidic conditions [22,24]. Future work should now investigate the impact of gut microbiota on this class specifically, by optimizing the extraction conditions and incubation and/or from other matrices in which betalains are more abundant, as in the case of beet root.

## Figures and Tables

**Figure 1 molecules-27-07568-f001:**
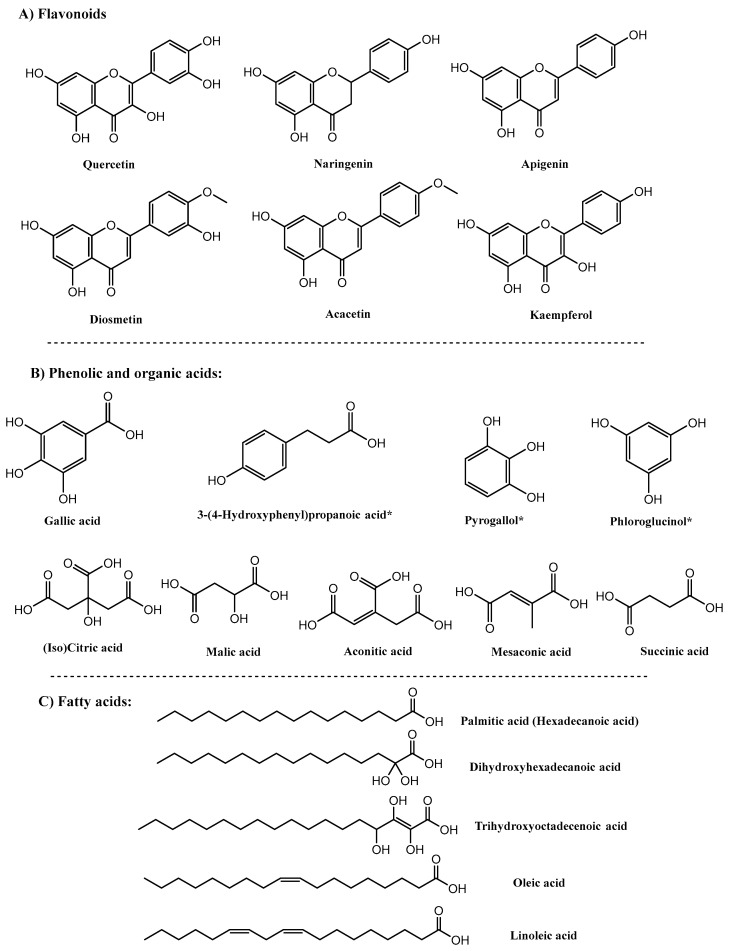
Chemical structures of detected metabolites from UHPLC-QTOF-MS analysis of *O. ficus* assayed in vitro and ex vivo with gut microbiota, * indicates biotransformed metabolites.

**Figure 2 molecules-27-07568-f002:**
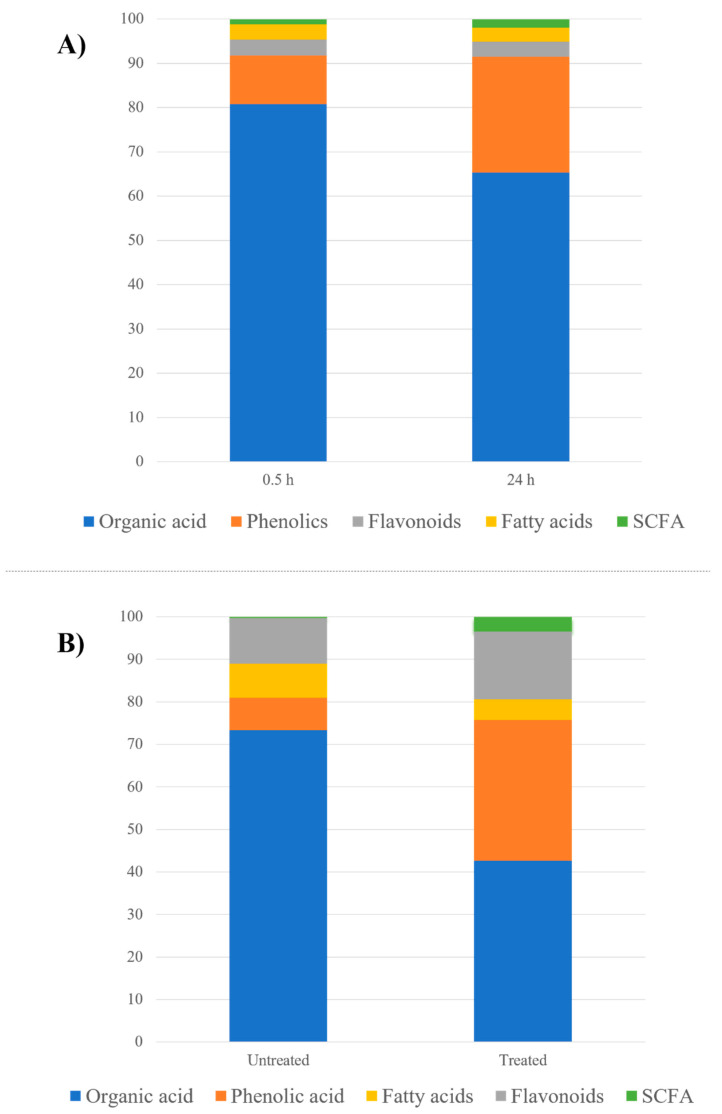
Bar chart showing changes in metabolite classes relative percentile levels set to 100% in *O. ficus* samples (**A**) treated in vitro at two time intervals, 0.5 and 24 h, with gut microbiota and (**B**) ex vivo untreated vs. treated samples with human fecal bacterial culture.

**Figure 3 molecules-27-07568-f003:**
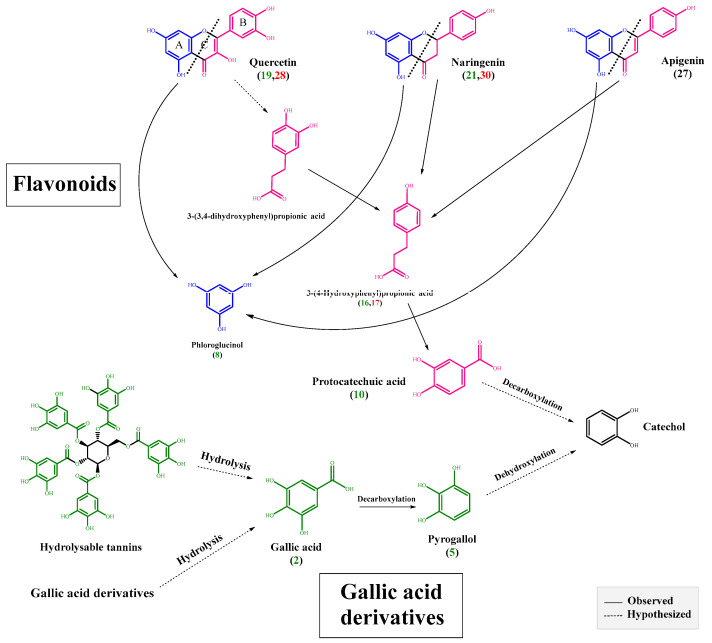
Proposed biotransformation pathways of flavonoids (red and blue) and gallic acid derivatives (green) in O. opuntia, (number) peak number of metabolites; green in in vitro assay and red in ex-vivo assay as presented in Table 1 and Appendix A.

**Figure 4 molecules-27-07568-f004:**
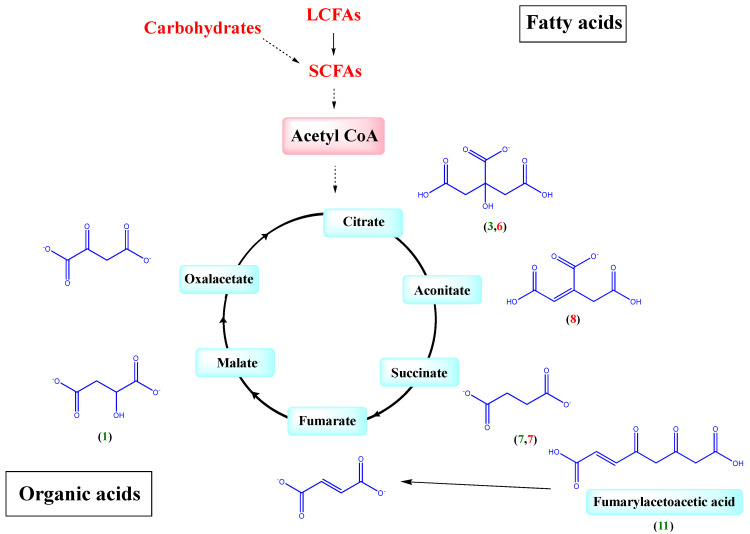
Proposed biotransformation pathways of fatty and organic acid mediated by the glyoxylate pathway, (number) peak number of metabolites; green in in vitro assay and red in ex vivo assay as present in Table 1 and Appendix A.

**Figure 5 molecules-27-07568-f005:**
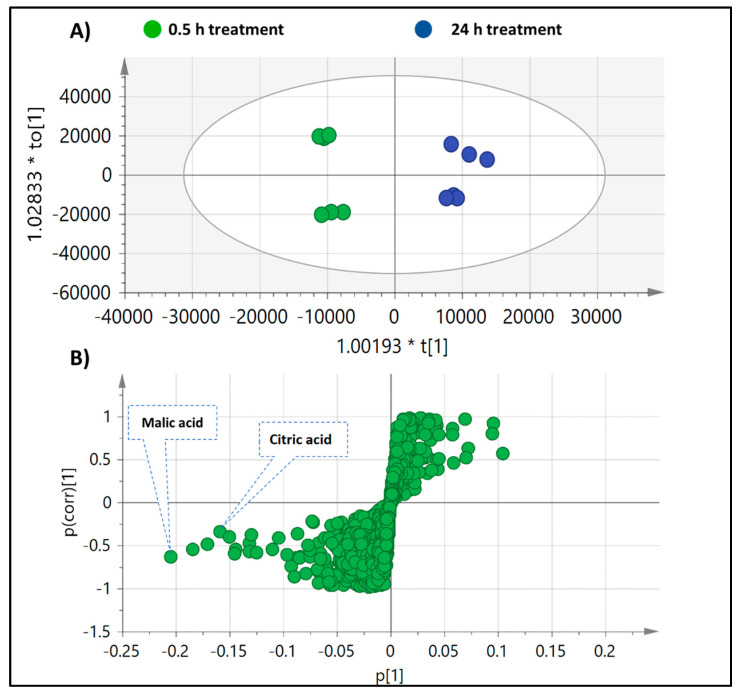
(**A**) OPLS model of *O. ficus* in vitro treated with gut microbiota based on incubation time; 24 h treatment samples (blue) modeled against 0.5 h (green). (**B**) S-plot of OPLS model, where metabolites with negative *p* values showed an increased abundance with 0.5 treatment samples, mainly malic and citric acid.

**Figure 6 molecules-27-07568-f006:**
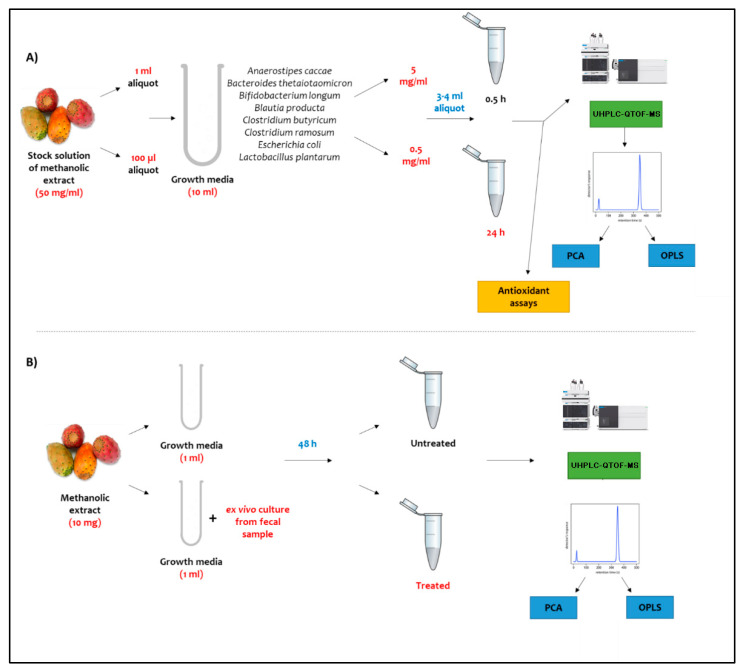
Schematic representation of the proposed (**A**) in vitro and (**B**) ex vivo protocol to assess the impact of gut microbiota on *O. ficus* metabolites.

**Table 1 molecules-27-07568-t001:** Metabolites identified using high-resolution UHPLC-QTOF-MS in *O. ficus* samples treated with gut microbiota at two time intervals, 0.5 and 24 h, along with their relative abundance.

Peak No.	[M-H]^−^	Rt _(sec)_	Molecular Formula	Error (ppm)	MS/MS	Name	Class	*O. ficus* Treated with Gut Microbiota (0.5 h) *	*O. ficus* Treated with Gut Microbiota (24 h) *
**1**	133.0160	66	C_4_H_6_O_5_	−9.08	115, 71.01	Malic acid	Organic acid	+	−
**2**	169.0158	79	C_7_H_6_O_5_	−11.61	125.02, 107.01, 97.03, 79.02	Gallic acid	Phenolic acid	+	++
**3**	191.0222	81	C_6_H_8_O_7_	−9.04	173.03, 129.01, 111, 99, 83.01	(iso)citric acid	Organic acid	+	−
**4**	207.0159	112	C_6_H_8_O_8_	−5.73	191.05, 127, 115, 99, 87, 73.03	Hydroxycitric acid	Organic acid	++	+
**5**	125.0256	139	C_6_H_6_O_3_	−9.21	107.01, 97.02, 79.02	Pyrogallol	Phenolics	−	+
**6**	117.0205	164	C_4_H_6_O_4_	−9.94	99.01, 73.03	Succinic acid	Organic acid	+	++
**7**	125.0256	171	C_6_H_6_O_3_	−9.66	107.01, 91.08, 79.01	Phloroglucinol	Phenolics	−	+
**8**	205.0368	174	C_7_H_10_O_7_	−7.28	191.05, 127, 111.01	Homocitric acid	Organic acid	+	−
**9**	153.0214	339	C_7_H_6_O_4_	−15.21	109.02, 93.03, 82	Protocatechuic acid	Phenolic acid	+	−
**10**	199.0265	356	C_8_H_8_O_6_	−10.21	101.38	Fumarylacetoacetic acid (Maleylacetoacetic acid)	Organic acid	+	++
**11**	117.0566	383	C_5_H_10_O_3_	−5.15	99.02	Hydroxypentanoic acid (hydroxyvaleric acid)	SCFA	+	++
**12**	541.2307	471	C_26_H_38_O_12_	0.92	315.13	Isorhamnetin glycoside	Flavonoids	++	+
**13**	219.0532	509	C_8_H_12_O_7_	−8.11	191.05, 127, 111.01, 87.01	Dimethyl citrate	Organic acid	++	+
**14**	183.032	553	C_8_H_8_O_5_	−11.75	168, 124.01, 97.02, 78.01	Methyl gallate	Phenolic acid	++	+
**15**	165.0578	589	C_9_H_10_O_3_	−15.19	147.04, 119.05, 91.01	3-(4-Hydroxyphenyl) propanoic acid	Phenolic acid	−	+
**16**	285.043	644	C_15_H_10_O_6_	−9.12	268.03, 243.03, 195.04, 169.06, 151.03	Kaempferol	Flavonoids	−	+
**17**	563.1102	691	C_25_H_24_O_15_	−10.56	447.09, 301.03, 151	Quercetin glycoside	Flavonoids	++	+
**18**	349.0618	713	C_9_H_18_O_14_	2.8	197.04, 169.01, 125.05	Ethyl gallate derivative	Phenolic acid	+	−
**19**	301.0387	788	C_15_H_10_O_7_	−10.4	179.07, 151	Quercetin	Flavonoids	+	++
**20**	271.0627	803	C_15_H_12_O_5_	−5.18	253.15, 209.36, 177.37, 151.01, 119.04	Naringenin	Flavonoids	+	−
**21**	287.2249	835	C_16_H_32_O_4_	−8.02	271.02, 243.05, 133.01, 115	Dihydroxyhexadecanoic acid	Fatty acids	+	++
**22**	443.1753	844	C_17_H_32_O_13_	2.99	329.23, 133.01, 71.01	Trihydroxyoctadecenoic acid derivative	Fatty acids	++	+
**23**	329.2358	860	C_18_H_34_O_5_	−7.33	133.01, 71.01	Trihydroxyoctadecenoic acid	Fatty acids	−	+
**24**	663.2948	874	C_41_H_44_O_8_	5.15	547.28, 431.26, 287.23, 133.01, 115	Dihydroxyhexadecanoic acid derivative	Fatty acids	+	−
**25**	547.2805	888	C_26_H_44_O_12_	−8.15	519.26, 431.26, 287.22, 143.03, 133.01, 115	Dihydroxyhexadecanoic acid derivative	Fatty acids	++	+
**26**	269.0478	893	C_15_H_10_O_5_	−6.99	251.16, 225.04, 201.06, 151, 117.03	Apigenin	Flavonoids	+	++
**27**	299.0577	902	C_16_H_12_O_6_	−11.77	284.03, 248.08, 151	Diosmetin	Flavonoids	+	−
**28**	283.0643	921	C_16_H_12_O_5_	−9.88	268.04, 239.03, 211.04, 179.03, 151.01, 117.03	Acacetin	Flavonoids	+	−
**29**	277.1822	997	C_17_H_26_O_3_	−4.53	253.18, 223.06, 123	Panaxytriol	Fatty alcohol	−	+
**30**	483.3161	1028	C_23_H_48_O_10_	0.82	379.08, 321.39, 255.23, 237.05	Palmitic acid derivative	Fatty acids	+	−
**31**	239.0701	1033	C_15_H_12_O_3_	6.99	207.04, 197.36, 135.03	Hydroxyflavanone	Flavonoids	+	−
**32**	295.2301	1039	C_18_H_32_O_3_	−7.16	277.21, 251, 183.13	Hydroxylinoleic acid	Fatty acids	+	−
**33**	243.1984	1066	C_14_H_28_O_3_	−6.11	219.01, 171.27, 99.02	Hydroxytetradecanoic acid	Fatty acids	+	−
**34**	271.2278	1132	C_16_H_32_O_3_	−0.76	253.19, 225.22	Hydroxyhexadecanoic acid	Fatty acids	+	++
**35**	471.3509	1132	C_30_H_48_O_4_	−6.39	429.35, 359.09, 306.09	Hydroxybetulinic acid	Triterpenoid	+	−
**36**	253.2196	1192	C_16_H_30_O_2_	−8.38	235.23, 209.15	Palmitoleic acid	Fatty acids	+	−
**37**	279.2351	1222	C_18_H_32_O_2_	−7.06	237.09, 187.01	Linoleic acid	Fatty acids	+	−
**38**	255.2355	1247	C_16_H_32_O_2_	−10.98	237.25, 183.1	Palmitic acid	Fatty acids	+	++
**39**	281.2521	1258	C_18_H_34_O_2_	−10.85	237.03, 171.1	Oleic acid	Fatty acids	+	−

* ++, +, −; reflects the metabolite relative abundance as depicted from the peak abundance data extracted from MS-DIAL, (++) increased abundance, (+) present, (−) absent.

## Data Availability

Not applicable.

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
