# Peer review of "Evaluation of Antioxidant Activity and Biotransformation of Opuntia Ficus Fruit: The Effect of In Vitro and Ex Vivo Gut Microbiota Metabolism"

_molecules, 2022, doi:10.3390/molecules27217568_

Round 1

Author Response

please see attached file addressing all reviewer comments

Reviewer 2 Report

This paper report the evaluation of Antioxidant Activity and Biotransformation of Opuntia Ficus Fruit. The effect of in vitro & ex vivo Gut Microbiota Metabolism authors must go through this

1. In the inroduction section should also mention some of the synthestic compound as antoxidant and how there natural product is better by using of some of the referrence paper authors may include https://doi.org/10.5012/jkcs.2021.65.2.106

2. Compounds should also be numbered both in figures and text.

3. Mass should be rechececked and all characterzed data.

4. Check for english languages, grammar and spellings mistakes.

5. Rerefrences are not in proper format please correct 

6. Follow author for instruction strictly.

Author Response

(The authors gave the same response as above.)

Round 2

Author Response

Please see attached response to all issues

Reviewer 2 Report

Comments incorporated

Author Response

we thank our reviewer for accepting the paper in this current version with no more comments